# Clusterin Is a Prognostic Biomarker of Lower-Grade Gliomas and Is Associated with Immune Cell Infiltration

**DOI:** 10.3390/ijms241713413

**Published:** 2023-08-29

**Authors:** Xiaoyue Ren, Chao Chang, Teng Qi, Pengyu Yang, Yuanbo Wang, Xiaorui Zhou, Feng Guan, Xiang Li

**Affiliations:** 1Provincial Key Laboratory of Biotechnology, Institute of Hematology, School of Medicine, Northwest University, Xi’an 710069, China; xiaoyuedoctor@163.com (X.R.); 15319788256@163.com (C.C.); qiteng@stumail.nwu.edu.cn (T.Q.); yangpengyu2016@163.com (P.Y.); yuanbo-wang@outlook.com (Y.W.); z2245572845@126.com (X.Z.); 2Key Laboratory of Resource Biology and Biotechnology in Western China, Ministry of Education, Provincial Key Laboratory of Biotechnology, College of Life Sciences, Northwest University, Xi’an 710069, China; guanfeng@nwu.edu.cn; 3College of Life Sciences, Northwest University, 229 Taibai North Road, Xi’an 710069, China

**Keywords:** clusterin, glioma, immune infiltration, prognostic

## Abstract

Dysregulation of clusterin (CLU) has been demonstrated in many cancers and has been proposed as a regulator of carcinogenesis. However, the roles of CLU in gliomas remain unclear. The expression of CLU was assessed using TIMER2.0, GEPIA2, and R package 4.2.1 software, leveraging data from TCGA and/or GTEx databases. Survival analysis and Cox regression were employed to investigate the prognostic significance of CLU. Immune infiltration was evaluated utilizing TIMER2.0, ESTIMATE, and CIBERSORT. The findings reveal the dysregulated expression of CLU in many cancers, with a marked increase observed in glioblastoma and lower-grade glioma (LGG). High CLU expression indicated worse survival outcomes and was an independent risk factor for the prognosis in LGG patients. CLU was involved in immune status as evidenced by its strong correlations with immune and stromal scores and the infiltration levels of multiple immune cells. Additionally, CLU was co-expressed with multiple immune-related genes, and high CLU expression was associated with the activation of immune-related pathways, such as binding to the antigen/immunoglobulin receptor and aiding the cytokine and cytokine receptor interaction. In conclusion, CLU appears to play crucial roles in tumor immunity within gliomas, highlighting its potential as a biomarker or target in glioma immunotherapy.

## 1. Introduction

Gliomas are the most common malignant primary brain tumors, accounting for 80.9% of malignant tumors in the central nervous system [1]. Glioblastoma (GBM, WHO grade IV), comprising 59.2% of gliomas [1], represents the most aggressive subtype, characterized by a 5-year survival rate of just 9.8% [2]. In contrast, lower-grade glioma (LGG, WHO grades II and III) exhibits a comparatively more favorable survival prognosis when compared to GBM. The currently standard treatment for gliomas is still confined to surgical resection and adjuvant chemotherapy with temozolomide combined with radiotherapy [2]. Although the standard treatment has been demonstrated to improve the survival of patients, these improvements are limited; the recurrence and progression of the tumor are still ineluctable, and the prognosis of patients remains unsatisfactory [3]. Immunotherapy is an innovative approach to cancer treatment that has demonstrated success across various cancer types [4,5]. Although there are still several challenges for the successful establishment of immunotherapy for gliomas, a better understanding of the immune infiltrates in the tumor microenvironment (TME) of gliomas may contribute to the development of more refined immunotherapies for the treatment of gliomas [6,7].

Clusterin (CLU) is an omnipresent conserved glycoprotein commonly secreted by cells that has been described as a stress-activated, ATP-independent molecular chaperone involved in a wide variety of pathological and physiological processes [8,9]. CLU is considered to be a regulator of carcinogenesis [10]. Elevated levels of CLU are observed in a variety of cancers, which exhibit close correlations with the risk for developing several cancers [11]. In addition, increasing evidence has demonstrated that CLU modulates a variety of cellular events associated with cancers, such as cancer stemness, epithelial–mesenchymal transition, cell survival, and treatment resistance, thus mediating the progression of many cancers, including breast carcinoma, renal cell carcinoma, bladder cancer, prostate cancer, hepatocellular carcinoma, esophageal cancer, colorectal cancer, ovarian cancer, and lung cancer [8,12,13,14,15,16]. Moreover, CLU seems to modulate tumor progression and metastasis by mediating the components of the TME. For example, CLU modulates the chemotactic migration and polarization of tumor-associated macrophages (TAM) by regulating the secretion of chemotactic cytokines [17], and it modulates the recruitment of dendritic cells (DCs) by regulating the expression of chemokine CCL20 [18]. Such properties potentialize CLU to be a promising target in cancer therapy. In central nervous system tumors, CLU has been observed to exhibit high expression levels in pituitary adenomas compared to its expression in non-neoplastic adenohypophyses [19]. Additionally, it has been identified as a tumor suppressor in neuroblastomas [20]. However, the specific functions of CLU in gliomas have been scarcely explored.

To this end, we designed this study to conduct a comprehensive investigation on the roles of CLU in gliomas in terms of three major aspects. First, expression of CLU in gliomas and its correlations with clinicopathologic features and prognosis were investigated. Second, the associations between CLU expression and the infiltration of immune cells as well as immune-checkpoint levels in the TME were explored. Third, the associations between CLU expression and the functional pathways were further investigated to uncover the underlying molecular mechanism. This study will provide evidence for the roles of CLU in gliomas and its potential to be a prognostic biomarker or therapeutic target in gliomas.

## 2. Results

### 2.1. Expression Pattern of CLU in Pan-Cancers

To assess the potential of clusterin (CLU) as a therapeutic target, the expression pattern of CLU was initially investigated in various types of cancers, comparing tumor tissues to their corresponding matched normal tissues. Based on the results obtained from TIMER2.0, we found that the expression of CLU was significantly reduced in the tumor tissue of a majority of cancers compared to their matched normal controls, such as bladder urothelial carcinoma (BLCA), breast invasive carcinoma (BRCA), and colorectal (COAD and READ) and lung (LUSC and LUAD) cancers. In contrast, CLU exhibited elevated expression levels in kidney renal clear cell carcinoma (KIRC), kidney renal papillary cell carcinoma (KIRP), thyroid carcinoma (THCA), and GBM when compared to tissues from matched control samples (Figure 1A). 

The Gene Expression Profiling Interactive Analysis (GEPIA), which relies on tumor tissues from The Cancer Genome Atlas (TCGA) and normal tissues from the Genotype-Tissue Expression (GTEx) project, unveiled that the expression patterns of CLU exhibited tumor specificity. Among 33 different cancer types, its highest expression was observed in GBM and LGG (Figure 1B). Additionally, expression levels of CLU across 33 cancer types from TCGA database were further analyzed using R package, and similar results were observed with the findings of TIMER2.0 (Figure 1C). Expression of CLU was significantly reduced in tumor tissue in a majority of cancers, while it was significantly elevated in the tumor tissues of KIRC, KIRP, THCA, and GBM.

### 2.2. Expression Pattern of CLU in Gliomas

We further investigated the expression distributions of CLU in gliomas. According to the expression profiles in TCGA, we found that CLU expression was associated with the tumor grades of gliomas, with markedly higher expression in GBM than in LGG (Figure 2A). CLU expression had been demonstrated to increase in tumor tissue of GBM samples than normal tissue, Figure 1A–C, thus the expression of GLU in LGG were investigated primarily in the following analyses. Normal tissues of LGG samples were absent in TCGA database (Figure 2B), therefore expression of CLU in LGG was analyzed based on the tumor tissues in TCGA and the normal tissue in GTEx, and elevated expression of CLU in tumor samples than normal samples was observed (Figure 2C). GEPIA analysis revealed consistent results that CLU was highly expressed in tumor samples than normal samples in LGG (Figure 2D). Protein expression of CLU in LGG and normal tissue samples were further verified using immunohistochemical staining (Figure 2E). Consistently, protein expression of CLU was markedly higher in LGG tissue samples in comparison with that of normal tissue samples (Figure 2F). These findings indicated a significantly elevated CLU expression at both mRNA and protein levels in LGG tissue samples.

### 2.3. CLU Expression Independently Associated with Prognosis of Patients with LGG

Prognostic value of CLU in LGG was explored using survival analyses on four survival outcomes. LGG patients with high CLU expression had significantly worse overall survival (OS, *p* < 0.001), disease-specific survival (DSS, *p* < 0.001), and progression-free interval (PFI, *p* < 0.001) in comparison with patients with low CLU expression (Figure 3A–C), indicating the associations of CLU expression with survival outcomes of LGG patients. However, no significant associations between CLU expression and disease-free interval (DFI, *p* = 0.342) of LGG patients were observed (Figure 3D). The independent force of CLU expression and three clinical factors were further explored. Forest plots revealed that both CLU expression and tumor grades were independently associated with the prognosis of LGG patients (Figure 3E,F). High CLU expression (hazard ratio of 1.451) and high glioma grades (hazard ratio of 3.145) were identified as two risk factors for prognosis of LGG patients.

### 2.4. Association of CLU Expression with Immune Infiltrates in LGG and GBM

The involvements of CLU in immune infiltrates in TME of LGG and GBM were further investigated using different methods. TIMER2.0 was applied to explore the correlations of CLU expression with the infiltration fractions of six main types of immune cells. LGG patients with high infiltration abundance of these six immune cells seemed to have worse cumulative survival in comparison with those patients with low infiltration abundance (Figure 4A). In addition to CD8 T cells, CLU expression showed outstanding positive correlations with the other five immune cells, including DCs (r = 0.341, *p* = 2.03 × 10^−14^), B cells (r = 0.183, *p* = 5.56 × 10^−5^), neutrophils (r = 0.328, *p* = 2.37 × 10^−13^), CD4+ T cells (r = 0.342, *p* = 1.54 × 10^−14^), and macrophages (r = 0.401, *p* = 1.15 × 10^−19^), which determined a negative correlation (r = −0.341, *p* = 1.54 × 10^−14^) of CLU expression with tumor purity in LGG (Figure 4B). Additionally, prominent positive correlations of CLU expression with stromal (r = 0.51, *p* < 2.2 × 10^−16^) and immune (r = 0.5, *p* < 2.2 × 10^−16^) scores were observed in LGG (Figure 4C,D). Moreover, correlation analysis of CLU expression with infiltration abundance of ten immune cells types inferred using CIBERSORT was further calculated (Figure 4E–N), and the results indicated that CLU expression positively correlated with CD8 T cells (r = 0.18, *p* = 0.00083), resting memory CD4 T cells (r = 0.21, *p* = 0.00012), macrophages M1 (r = 0.14, *p* = 0.0096), and resting mast cells (r = 0.15, *p* = 0.0068), while negatively correlated with activated mast cells (r = −0.15, *p* = 0.0051) in LGG.

In terms of GBM, infiltration abundance of six immune cells in TIMER2.0 showed no significant associations with cumulative survival of patients, in addition to DCs (Figure 5A). Similar to the findings in LGG, CLU expression showed outstanding positive correlations with five immune cells (all *p* < 0.05), except CD8 T cells in TIMER2.0 analysis (Figure 5B), and showed prominent positive correlations with stromal and immune scores (all r > 0.3, *p* < 0.05, Figure 5C,D). Additionally, CLU expression exhibited positive correlations with monocytes (r = 0.3, *p* = 0.00012) and resting memory CD4 T cells (r = 0.26, *p* = 0.00069), while negative correlations with macrophage M0 (r = −0.17, *p* = 0.031) and M2 (r = −0.33, *p* = 1.7 × 10^−5^) in GBM, which were inconsistent with the findings in LGG, and the detailed comparison of various parameters between LGG and GBM is shown in Table 1 and Table 2.

### 2.5. Associations of CLU Expression with Immune Status-Related Genes

To further uncover the possible mechanisms of CLU involved in immune infiltration, associations of CLU expression with several groups of immune-related genes were explored. There were prominent positive correlations between CLU expression and expression of almost all immunosuppressive genes and most immune-activated genes, such as CD274, CTLA4, and PDCD1LG2 (Figure 6A,B). Moreover, CLU expression positively correlated with several chemokine ligands and receptors, such as CCR1/2/3/4/5, CXCR2/3/4, and CCL3/4/5, whereas negatively correlated with chemokine ligands and receptors, such as CCR6, CXCR5, and CCL1 (Figure 6C,D). We further investigated the expression of eight immune-checkpoints (Figure 6E) and found that expression of all these immune-checkpoints (e.g., CD274, PDCD1, and CTLA4, all *p* < 0.05) were markedly elevated in LGG patients with high CLU expression in comparison to SIGLEC15 (*p* = 0.43). These findings highlighted the involvements of CLU in immune status and its potential value in immunotherapy.

### 2.6. CLU Expression-Associated Functional Pathways

Gene set enrichment analysis (GSEA) was applied to investigate the significantly functional pathways associated with CLU expression. Several immune-related gene ontology (GO) terms, such as immunoglobulin complex, phagocytosis recognition, binding to antigen/immunoglobulin receptor, were found to be associated with high CLU expression (Figure 7A). Additionally, immune-related Kyoto Encyclopedia of Genes and Genomes (KEGG) pathways, such as primary immunodeficiency and cytokine and cytokine receptor interaction, were activated with high CLU expression (Figure 7B). These results might uncover the potential mechanism of CLU involvement in immune status.

## 3. Discussion

Gliomas are the most common type of malignant intracranial tumors with extremely worse prognosis. Even with standard treatment, the recurrence and progression of tumor are still ineluctable for patients [21]. Increasing attention has been paid to prognosis-associated molecular markers in gliomas, such as 1p19q co-deletion, IDH mutations, and p53 mutation, as their application in clinical practice for gliomas has been demonstrated [22]. In addition, promising novel therapies, such as gene therapy and immunotherapy, are constantly being established [23,24]. Thus, it is crucial to identify additional prognostic biomarkers and develop new treatment strategy to optimize treatment in gliomas.

CLU has gained increasing attention due to its paradoxical and multifunctional properties in various pathologies [9] and is regarded as a regulator of carcinogenesis and a therapeutic target in cancers [8,10]. However, expression and the specific roles of CLU in glioma are rarely investigated. To this end, we conducted this study to conduct a comprehensive investigation on the roles of CLU in gliomas.

Expression of CLU in pan-cancers was first investigated, and it was found that CLU was down-regulated in most cancers. This seemed to be inconsistent with the previous reports that expression of CLU was elevated in many cancers [8,11], and this might be explained partly by the sample difference and the paradoxical property of CLU. The differential expression of CLU in a variety of cancers highlighted the importance of CLU in cancers, and the potential of CLU as a therapeutic target. Among 33 types of cancers, CLU showed highest expression in gliomas, including both GBM and LGG. In gliomas, CLU expression was markedly elevated in tumor tissue than in normal tissue, and immunohistochemical staining of CLU in clinical samples verified the high expression of CLU in tumor tissue. High CLU expression was found to be associated with worse survival outcomes. We found that CLU expression was associated with glioma grades, with higher expression levels in GBM than in LGG. Further Cox regression analyses indicated that high CLU expression was an independent risk factor for prognosis of LGG patients. Such findings indirectly explained why LGG patients with high CLU expression had a worse survival outcome. Effective prognostic biomarkers are important for the clinical management and making treatment decisions for patients because they provide key information in terms of tumor progression or clinical outcome [25,26]. Our analyses demonstrated that CLU was an independent prognostic biomarker in gliomas, associated with glioma grades and worse survival outcomes in gliomas.

Diverse immunotherapies have been developed in gliomas; however, unlike its successful use in other tumors, the effects of several immunotherapy strategies appear to be limited in gliomas [24]. The effect of immunotherapy relies mainly on the infiltrating immune cells within the TME of tumor, which are the crucial part of TME that can modulate the progression of tumors by their dynamic and extensive crosstalk with tumor cells [27,28]. Several molecules have been demonstrated to participate in such intercellular crosstalk [29,30]. Results from our analyses indicated that high CLU expression positively correlated with the infiltrating abundance of different immune cells, such as macrophages, DCs, and CD4+ T cells, which suggest that CLU might modulate the recruitment of immune cells in TME of gliomas to some extent. We further found that high CLU expression strongly correlated with multiple immune checkpoints and immune status-related genes as well as chemokine-related genes. Consistently, studies have reported that CLU modulates the chemotactic migration and polarization of TAM by regulating the secretion of chemotactic cytokines [17], and modulates the recruitment of DCs by regulating expression of chemokine CCL20 [18]. In addition, Yang et al. revealed that CLU was involved in the recruitment of immune cells in breast tumors, and its elevated expression was closely related to multiple specific immune cell subset-associated molecular markers [31]. Additionally, GSEA indicated that multiple immune-related biological processes and pathways were activated in LGG patients with high CLU expression. All these findings emphasized the close involvement of CLU in immune status of gliomas. Thus, we speculated that CLU might have the potential to be a biomarker or a specific target in glioma immunotherapy. Even though this study is preliminary, it still has several limitations about how CLU can be adapted and modulated with immunotherapy. Then, more mechanical and functional experiments should be performed to explore the effects of CLU on the immune cells within the tumor microenvironment in gliomas, in order to develop personalized approaches such as vaccination and explore multiple clinical trials investigating immunotherapy combination studies.

## 4. Materials and Methods

### 4.1. Data Sources and Differential Analysis of CLU Expression

Expression of CLU between tumor tissues and the corresponding normal tissues in pan-cancer were analyzed using the “Gene_DE Module” of TIMER2.0 (http://timer.cistrome.org/ 2022 accessed on 14 August 2023) [32] and GEPIA (version 2, http://gepia.cancer-pku.cn/ 2022 accessed on 14 August 2023) [33]. In addition, the transcriptome data of 33 pan-cancer types in the TCGA (https://tcga.xenahubs.net 2022 accessed on 14 August 2023) [34] were downloaded, and the differential expression of CLU was analyzed with the aid of Wilcoxon test provided in R package. Through these three ways, data were divided into high and low expression groups based on the median expression value of CLU expression. False Discovery Rate < 0.05 indicated statistical significance. The abbreviations and full names of the these tumor types are shown in Abbreviation Table.

Genes expression profiles of both LGG and GBM in TCGA, and genes expression profiles of normal samples in the GTEx (http://commonfund.nih.gov/GTEx/, accessed on 14 August 2023) [35] database were downloaded to conduct the differential analysis of CLU expression between tumor and normal tissues using Wilcoxon test. Verification of the differential expressions of CLU in LGG and normal tissues was conducted using GEPIA. FDR < 0.05 indicated statistical significance. Expression distributions of CLU were visualized in the form of a boxplot with the aid of “ggpubr” R package.

### 4.2. Immunohistochemical Staining

LGG (n = 10) and normal (n = 10) tissue samples were collected at the First Affiliated Hospital of Yan’an University from November 2020 to April 2023. The clinical characteristics of patients and healthy controls are detailed in Table 3. All LGG tissue samples were confirmed by pathologists. Protein expression of CLU in these tissue samples were analyzed utilizing immunohistochemical staining. In short, the tissues were firstly prepared as 3.5 µm-thick paraffin sections. Following deparaffinization, rehydration, and antigen retrieval, the sections were incubated with anti-CLU (SC-166907, 1:500, Santa Cruz, CA, USA) at 4 °C overnight. All slides were covered with poly-HRR goat anti-body and counter-stained with diaminobenzidine solution (3–5 min) and hematoxylin. Finally, the section was examined under a fluorescent microscope and analyzed using Image-Pro plus 6.0 (Media Cybernetics, Inc., Rockville, Maryland). The study was approved by the Research Ethics Committee of Northwest University, and all participants provided informed consent in accordance with the Declaration of Helsinki.

### 4.3. Association between CLU Expression and Prognosis in LGG

The clinicopathological and survival data of LGG samples were downloaded from TCGA database. To assess the prognostic value of CLU in LGG, all the LGG samples were assigned into high CLU expression and low CLU expression on the basis of their median expression values. Then, survival analysis, including OS, DSS, PFI, and DFI, were assessed utilizing R survival package and were visualized as Kaplan–Meier curves with log-rank *p*-values. Additionally, the independent prognostic value of CLU and clinical factors, including age, gender, and tumor grades, were assessed using univariate and multivariate Cox regression analyses. The results were displayed in the form of forest plots using “forestplot” R package.

### 4.4. Immune Infiltration in LGG and GBM

The correlation between CLU and immune cell infiltration was assessed using the “Gene Module” of TIMER2.0 [32], which generated scatter plots of Spearman’s correlations between CLU expression and tumor purity as well as the abundance of six immune cells types, including DCs, B cells, neutrophils, CD4+ T cells, macrophages, and CD8+ T cells. Stromal and immune scores were calculated using R “ESTIMATE” package, and the sum of these two score was the ESTIMATE score, which could indirectly reflect the tumor purity [36]. The infiltrating abundance of immune cells in LGG and GBM tissues were estimated using the CIBERSORT algorithm, a computational method for quantifying fractions of 22 cell types from bulk tissue gene expression profiles. The correlations of CLU expression with stromal and immune scores as well as infiltration abundance of immune cells were further analyzed using R packages “ggplot2,” “ggpubr,” and “ggExtra”.

### 4.5. Co-Expression Analysis and Immune-Checkpoint Analysis in LGG

Co-expression analysis between CLU expression and the expression of immunosuppressive and immune activation genes and chemokine and chemokine receptor-related genes was conducted using the R “limma” package. Expression data of immune-checkpoint genes were extracted, and their expression in two LGG groups stratified by high and low CLU expression was analyzed utilizing R “ggplot2” package.

### 4.6. Gene Set Enrichment Analysis (GSEA)

With the predefined GO and KEGG gene sets in GSEA website as an enrichment reference, GSEA was conducted to analyze the GO annotations terms and KEGG pathways that were significantly associated with CLU expression using the R package “enrichplot”.

## 5. Conclusions

In conclusion, this was the first study to investigate the expression and specific roles of CLU in gliomas. This study provided evidences that CLU was highly expressed in gliomas and might be an independent indicator to predict worse survival outcomes for LGG patients. We confirmed the close involvement of CLU in immune status of gliomas, which could be used as a biomarker or a specific target in glioma immunotherapy. Further investigations focusing on the roles of CLU in gliomas are needed in future.

## Figures and Tables

**Figure 1 ijms-24-13413-f001:**
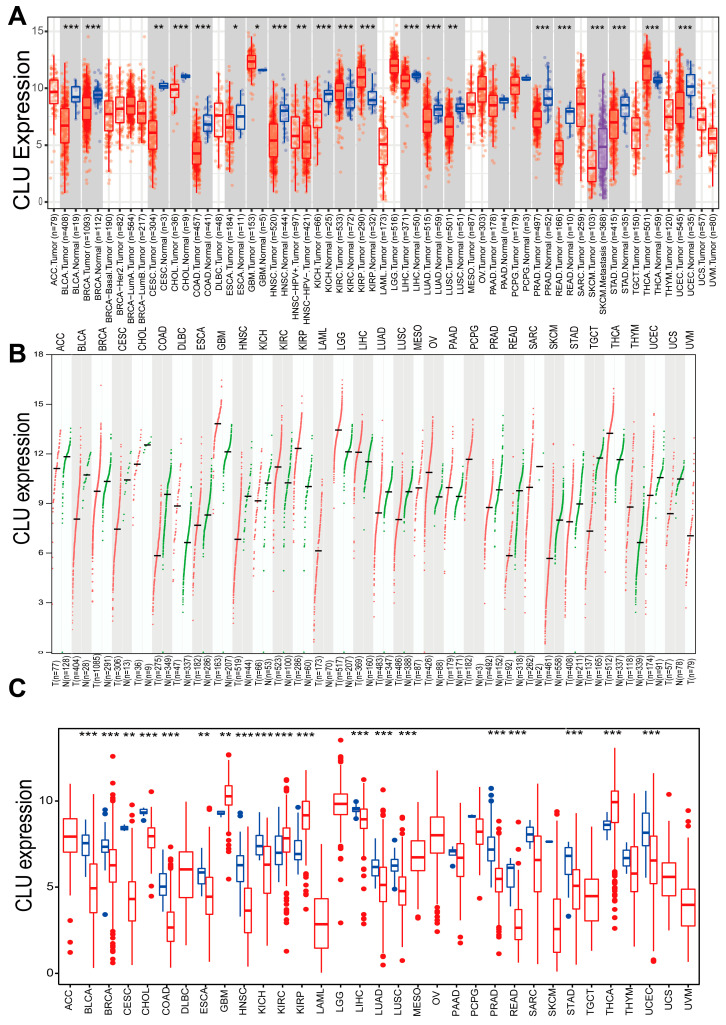
Expression of CLU in pan-cancers. Expression levels of CLU in distinct cancers or tumor tissues and neighboring non-tumor tissues analyzed using TIMER2.0 ((**A**), red boxes represent tumor tissues, and green boxes represent normal tissues). GEPIA2 ((**B**), each dot represents the expression profile in one sample) and R package based on TCGA and/or GTEx databases (**C**). * *p* < 0.05, ** *p* < 0.01, *** *p* < 0.001.

**Figure 2 ijms-24-13413-f002:**
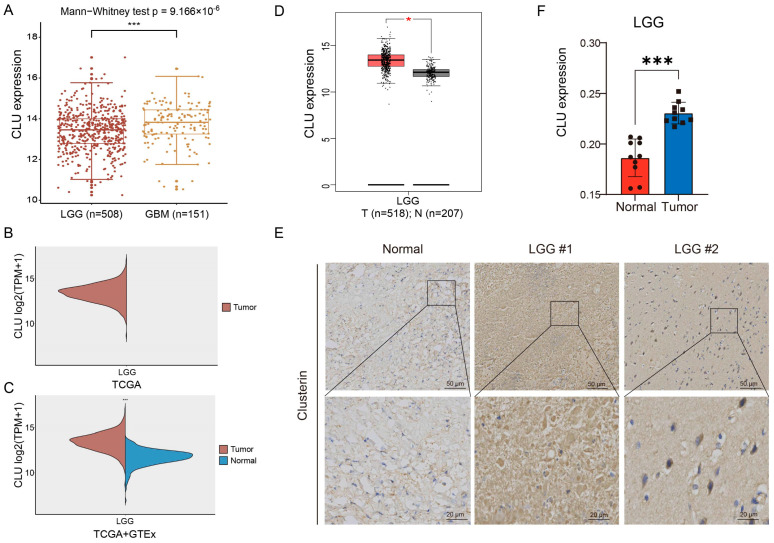
Expression pattern of CLU in gliomas. (**A**) Differential CLU expression between LGG (WHO grade II and III) and GBM (WHO grade IV). (**B**) Levels of CLU in LGG tissue samples in TCGA. (**C**) Differential expression of CLU between LGG tissue in TCGA and normal tissue samples in GTEx. (**D**) Expression levels of CLU in between LGG tissue and normal tissue analyzed using GEPIA2. (**E**,**F**) Representative images and quantification of immunohistochemical staining for CLU in LGG tissue and normal tissue. * *p* < 0.05, *** *p* < 0.001.

**Figure 3 ijms-24-13413-f003:**
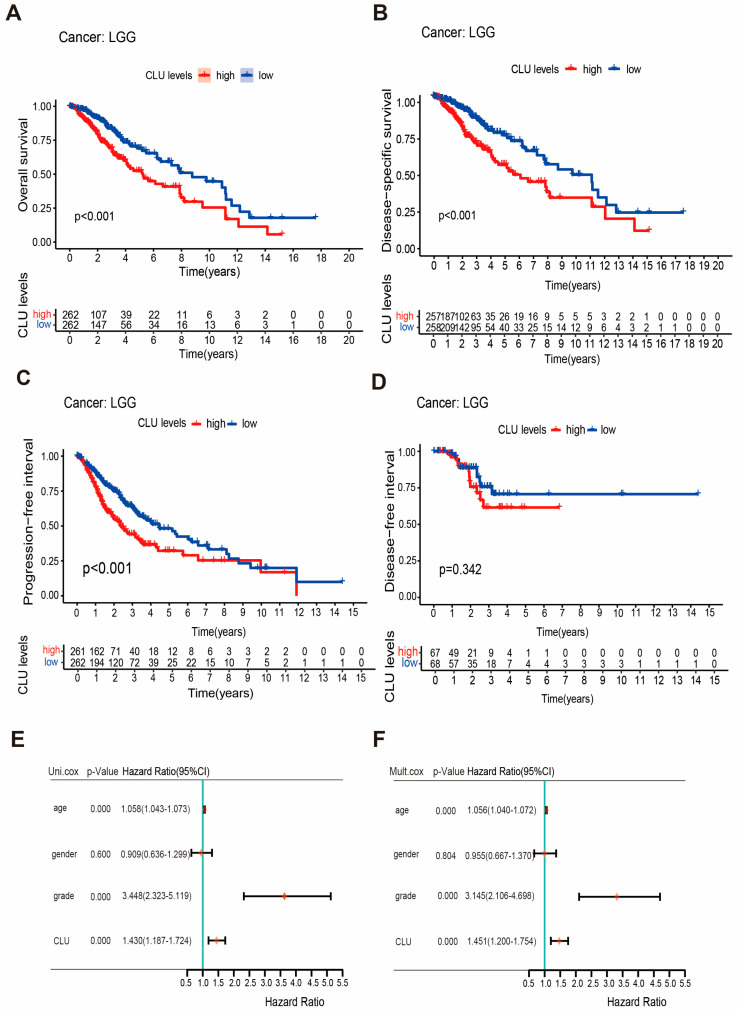
Prognostic value of CLU in LGG. (**A**–**D**) Survival curves showing the differences on overall survival (**A**), disease-specific survival (**B**), progression-free interval (**C**), and disease-free interval (**D**) between high and low CLU expression; (**E**,**F**), forest plots showing the results of univariable (**E**) and multivariate (**F**) Cox regression.

**Figure 4 ijms-24-13413-f004:**
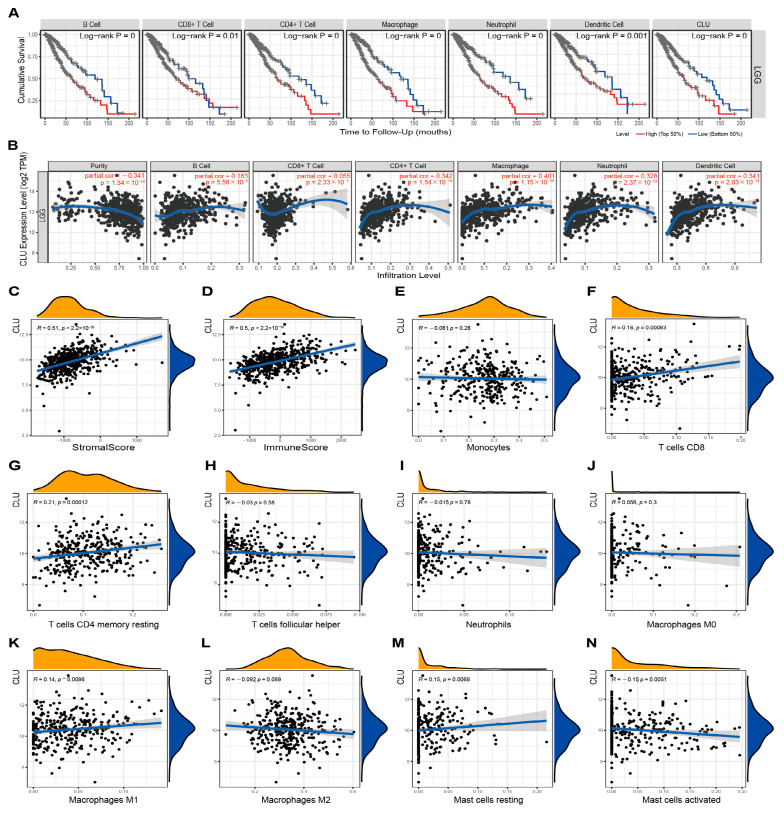
Associations of CLU with immune infiltration in LGG. (**A**) Cumulative survival of LGG patients with high and low infiltrating levels of six immune cells in TIMER2.0. (**B**) Correlations between CLU expression and infiltrating levels of six immune cells in TIMER2.0. (**C**,**D**) Correlations of CLU expression with stromal (**C**) and immune (**D**) scores. (**E**–**N**) Correlations between CLU expression and infiltrating levels of immune cells using CIBERSORT.

**Figure 5 ijms-24-13413-f005:**
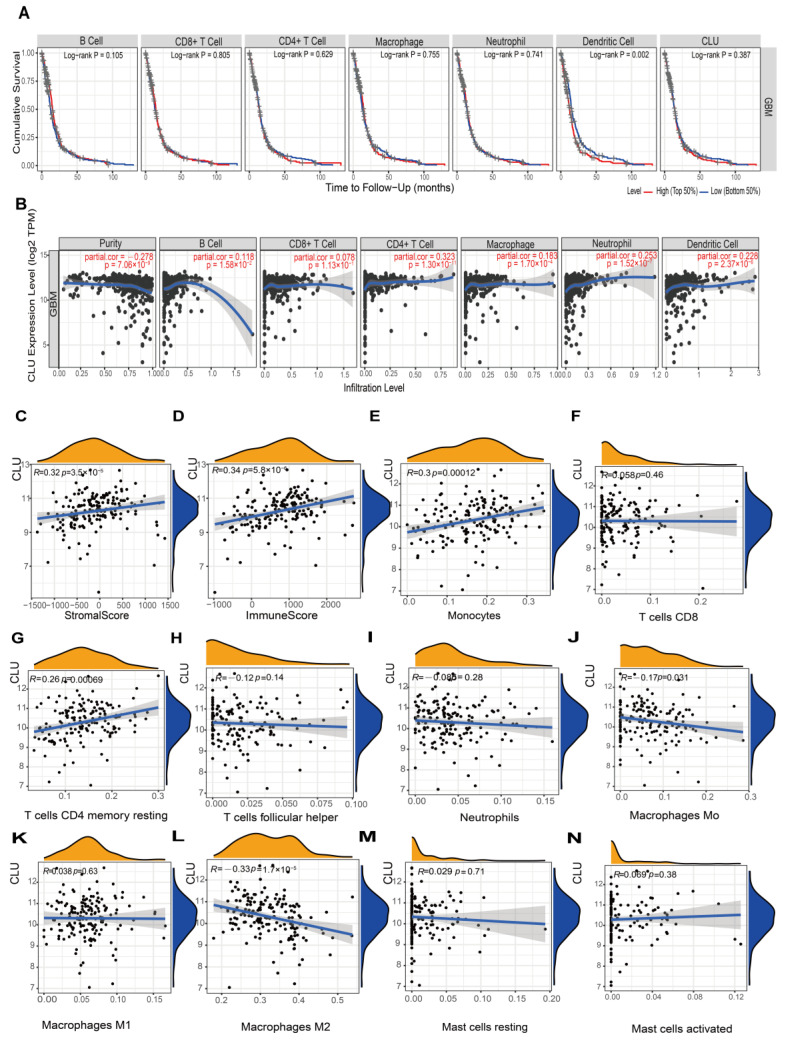
Associations of CLU with immune infiltration in GBM. (**A**) Cumulative survival of GBM patients with high and low infiltrating levels of six immune cells in TIMER2.0. (**B**) Correlations between CLU expression and infiltrating levels of six immune cells in TIMER2.0. (**C**,**D**) Correlations of CLU expression with stromal (**C**) and immune (**D**) scores. (**E**–**N**) Correlations between CLU expression and infiltrating levels of the immune cells inferred using CIBERSORT.

**Figure 6 ijms-24-13413-f006:**
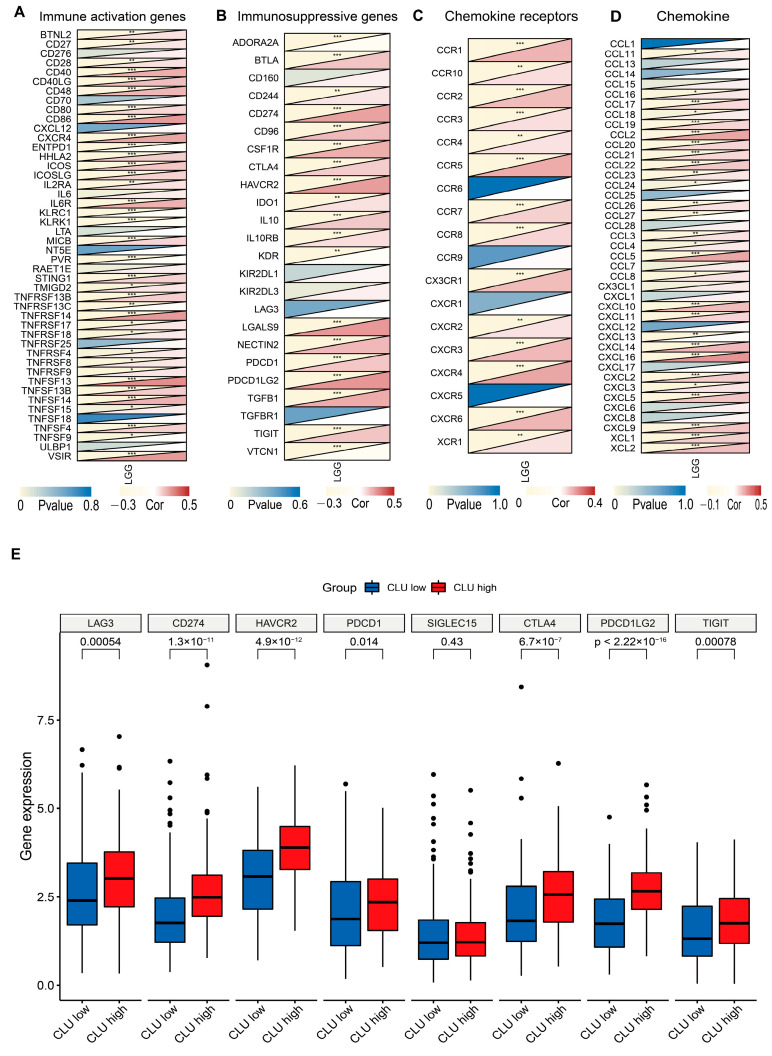
Associations of CLU expression with immune status-related genes. (**A**–**D**) Co-expression of CLU expression with immune activation (**A**), immunosuppressive (**B**), chemokine receptors (**C**), and chemokine (**D**) genes. (**E**) Boxplots showing the differential expression of immune-checkpoints in high and low CLU expression groups. * *p* < 0.05, ** *p* < 0.01, *** *p* < 0.001.

**Figure 7 ijms-24-13413-f007:**
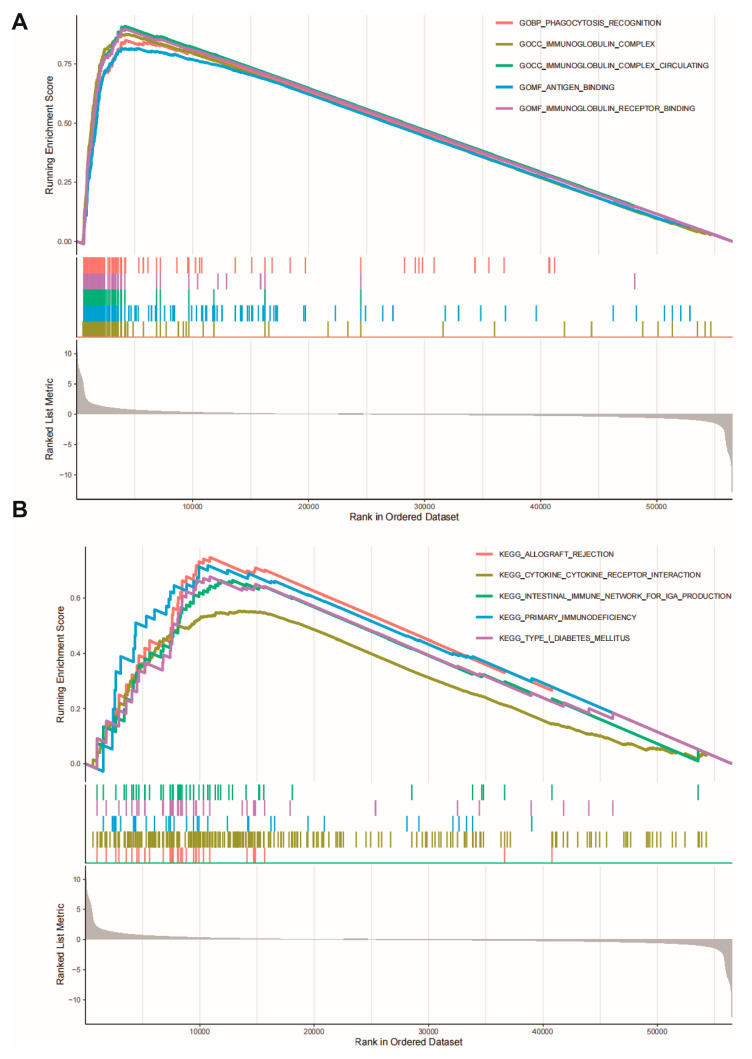
Gene set enrichment analysis. CLU expression significantly associated with gene ontology terms (**A**) and KEGG pathways (**B**).

**Table 1 ijms-24-13413-t001:** Correlation analysis between CLU and immune cells using CIBERSORT in LGG and GBM.

Description	LGG		GBM	
	R	*p*	R	*p*
Stromal Score	0.51	**2.20 × 10^−16^**	0.32	**3.50 × 10^−5^**
Immune Score	0.5	**2.20 × 10^−16^**	0.34	**5.80 × 10^−6^**
Monocytes	−0.061	2.60 × 10^−1^	0.3	**1.20 × 10^−4^**
T cell CD8	0.18	**8.30 × 10^−4^**	0.058	4.60 × 10^−1^
T cell CD4 memory resting	0.21	**1.20 × 10^−4^**	0.26	**6.90 × 10^−4^**
T cells follicular helper	−0.03	5.80 × 10^−1^	−0.12	1.40 × 10^−1^
Neutrophils	−0.015	7.80 × 10^−1^	−0.085	2.80 × 10^−1^
Macrophages M0	0.056	3.00 × 10^−1^	−0.17	**3.10 × 10^−2^**
Macrophages M1	0.14	**9.60 × 10^−3^**	0.038	6.30 × 10^−1^
Macrophages M2	−0.092	8.90 × 10^−2^	−0.33	**1.70 × 10^−5^**
Mast cells resting	0.15	**6.80 × 10^−3^**	0.029	7.10 × 10^−1^
Mast cells activated	−0.15	**5.10 × 10^−3^**	0.069	3.80 × 10^−1^

Bold fonts represent *p* < 0.05.

**Table 2 ijms-24-13413-t002:** Correlation analysis between CLU and immune cells using TIMER in LGG and GBM.

Description	LGG		GBM	
	cor	*p*	cor	*p*
Purity	−0.341	**1.54 × 10^−14^**	−0.278	**7.06 × 10^−9^**
B Cell	0.183	**5.56 × 10^−5^**	0.118	**1.58 × 10^−2^**
CD8+ T Cell	0.055	2.33 × 10^−1^	0.078	1.13 × 10^−1^
CD4+ T Cell	0.342	**1.54 × 10^−14^**	0.323	**1.30 × 10^−11^**
Macrophage	0.401	**1.15 × 10^−19^**	0.183	**1.70 × 10^−4^**
Neutrophil	0.328	**2.37 × 10^−13^**	0.253	**1.52 × 10^−7^**

Bold fonts represent *p* < 0.05.

**Table 3 ijms-24-13413-t003:** Clinical characteristics of LGG patients and healthy controls for validation.

Diagnosis	Age (Year)	Gender	WHO Classification	Location of the Histological Lesion
LGG	32	F	WHO II	right cingulated gyrus
LGG	39	F	WHO I	right frontal parietal lobe
LGG	67	M	WHO II	left temporal lobe
LGG	35	M	WHO I	left temporal lobe
LGG	50	M	WHO II	left frontotemporal Lobe
LGG	55	M	WHO IV	left temporal lobe
LGG	31	M	WHO II	left temporoparietal occipital lobe
LGG	56	M	WHO II	right frontal insular lobe
LGG	48	M	WHO II	right parietal lobe
LGG	47	F	WHO II	intraspinal
HD	36	M	N	N
HD	30	F	N	N
HD	38	F	N	N
HD	39	F	N	N
HD	34	F	N	N
HD	32	F	N	N
HD	30	F	N	N
HD	34	F	N	N
HD	34	F	N	N
HD	32	F	N	N

## Data Availability

Publicly available datasets were used in this study. The TCGA gene expressions along with the clinical datasets were downloaded from the TCGA hub (https://tcga.xenahubs.net accessed on 14 August 2023) and GTEx database (http://commonfund.nih.gov/GTEx/ accessed on 14 August 2023).

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
