# Peer review of "Clusterin Is a Prognostic Biomarker of Lower-Grade Gliomas and Is Associated with Immune Cell Infiltration"

_ijms, 2023, doi:10.3390/ijms241713413_

Round 1
Reviewer 1 Report
Here X. Ren et al demonstrated Clusterin (CLU) as potential target for Glioma. The authors utilized various expression analysis tools and showed marked increase of CLU in Glioma (low grade LGG & glioblastoma GBM). Additionally, it was correlated with worse survival outcomes and can be used for patient prognosis. In terms of mechanism, the authors correlate CLU expression with immune response, including antigen receptor binding and cytokine/receptor interactions.
Considering the poor outcome of Glioma patients, finding biomarkers and potential therapeutic targets are critical. Coupled with the extensive results presented, the article warrants significant interests and is of significant quality to be published in IJMS, subject to several revisions/clarifications as below:
- For Fig 2, CLU in 2E is marked by brown color staining. In this case, what does the dark blue spot represent? Moreover, what is the different between LGG#1 & #2, as CLU expression is quite distinct between the two samples.
- For Fig 3, the authors categorized CLU expression into high / low for correlation to patient outcomes. Please explain the threshold utilized for categorizing such CLU expression. Was this based on mRNA or protein expression?
- In Fig 6, the authors showed that CLU expression positively correlated with immuno-suppressive genes as well as immune activated genes. Please elaborate how this can be explained. Should it not only suppress or activate immune responses?
- The authors presented the potential of CLU as biomarker for glioma well, but there is minimal description for authors' claim to utilize CLU as specific target in gliomas immunotherapy. Please elaborate on how CLU can be adapted and modulated prior or in conjunction to immunotherapy.
Author Response
Dear Reviewer, Thank you very much for your comments and suggestions. Those comments are all valuable and very helpful for revising and improving our paper, as well as the important guiding significance to our researches. Please see the attachment about the point-by-point response.
Reviewer 2 Report
In the paper : Clusterin is a prognostic biomarker in Gliomas and associated with immune cell infiltrationThe author looked at how clusterin functions in various types of cancer, concentrating particularly on how it functions in gliomas and how it relates to inflammatory markers. They show that clusterin (CLU) is dysregulated in several cancers and have suggested that it is a regulator of carcinogenesis. The author goes into detail on three topics in particular: the prognosis, clinicopathological features, and the expression of CLU in gliomas. Second, relationships between CLU expression and immune cell infiltration as well as levels of immunological checkpoints in TME were investigated. Third, in order to understand the underlying molecular mechanism, the relationships between CLU expression and functional pathways were further examined.
Despite the excellent writing and organization of the report, I have the following questions:
1) In line 102 par. 2.2, the authors investigated the expression of Clu in 10 samples of LGG using immunohistochemistry; however, they provided only minimal information regarding the samples' provenance, age, and the location of the histological lesion,. The status of IDH1 mutation is not mentioned… I would suggest creating a table.
2) Could glioblastoma tissue be used for the same immunochemistry assays by the authors?
3)Could the authors clarify the number of samples considered illustrating the connection between ClU and immune infiltration in the Timer 2.0 gene module?
4) In the event that the writers do not include in-depth glioma research, it would be wise to change the title Clusterin is a prognostic biomarker of LGG Gliomas and is associated with immune cell infiltration
Moderate editing of English language required
Author Response

(The authors gave the same response as above.)

Reviewer 3 Report
Comments attached

Author Response

(The authors gave the same response as above.)

Round 2
Reviewer 3 Report
The authors have addressed the concerns appropriately by incorporating additional materials in the revised manuscript. By considering the author's limitations and by overall substantial improvement of the revised manuscript can be considered for publication.